# High-speed modulation of a terahertz quantum cascade laser by coherent acoustic phonon pulses

Aniela Dunn [1✉], Caroline Poyser[2], Paul Dean[1], Aleksandar Demić[1], Alexander Valavanis [1], Dragan Indjin[1], Mohammed Salih[1], Iman Kundu [1], Lianhe Li [1], Andrey Akimov[2], Alexander Giles Davies [1], Edmund Linfield [1], John Cunningham [1] & Anthony Kent[2]

The fast modulation of lasers is a fundamental requirement for applications in optical communications, high-resolution spectroscopy and metrology. In the terahertz-frequency range, the quantum-cascade laser (QCL) is a high-power source with the potential for high-frequency modulation. However, conventional electronic modulation is limited fundamentally by parasitic device impedance, and so alternative physical processes must be exploited to modulate the QCL gain on ultrafast timescales. Here, we demonstrate an alternative mechanism to modulate the emission from a QCL device, whereby optically-generated acoustic phonon pulses are used to perturb the QCL bandstructure, enabling fast amplitude modulation that can be controlled using the QCL drive current or strain pulse amplitude, to a maximum modulation depth of 6% in our experiment. We show that this modulation can be explained using perturbation theory analysis. While the modulation rise-time was limited to ~800 ps by our measurement system, theoretical considerations suggest considerably faster modulation could be possible.

[1] School of Electronic and Electrical Engineering, University of Leeds, Leeds LS2 9JT, UK. [2] School of Physics and Astronomy, University of Nottingham, Nottingham NG7 2RD, UK. ✉email: A.Dunn1@Leeds.ac.uk

Terahertz (THz) frequency technologies have undergone remarkable development over the past few decades, motivated in part by the diverse range of potential applications[1, 2]. Foremost among these developments has been the THz-frequency quantum-cascade laser (QCL)[3,4]—a compact, electrically pumped source based on intersubband transitions in a quantum heterostructure. QCLs exhibit ~ps carrier lifetimes, potentially allowing ultrafast amplitude or frequency modulation[5], a key requirement for frequency shifting, high data-bit transmission[6], active mode-locking[7] and frequency comb synthesis[8], as well as amplitude, frequency[9] and phase stabilisation[10]. As such, these sources are ideally suited for applications in THz metrology[11], high-resolution spectroscopy[12,13] and ultra-fast wireless communications[6,14].

The most common modulation method relies on direct gain control via the bias voltage[15]. Crucially, the upper-state lifetime (~ps) of the lasing transition in THz QCLs is typically shorter than the cavity round-trip time (~50 ps). As such, the laser power and gain do not exhibit relaxation oscillations, in principle permitting direct modulation up to ~100 GHz[5]. In practice, however, modulation bandwidths are limited by parasitic inductance and phase-matching considerations. This demands the use of coplanar probes or matched microstrip-lines for fast electrical modulation[16,17], which has allowed modulation frequencies up to 35 GHz to be demonstrated in QCLs with metal–metal waveguides. QCLs with surface-plasmon waveguides, though offering far superior beam quality[18], are only suitable for direct electrical modulation below ~20 GHz[19].

Other THz-QCL modulation schemes rely on controlling effective cavity losses. For example, graphene-based modulators allow 100% modulation in either an external-cavity configuration[20], albeit only close to the lasing threshold or monolithically integrated with the QCL[21]. Although >100 MHz modulation is achievable with the latter approach, the bandwidth is still fundamentally limited by the parasitic capacitance of the modulating element.

To overcome this limit, one must exploit alternative physical processes to control the QCL gain or cavity losses. To this end, optically controlled external-cavity metamaterial modulators enable a small degree of amplitude control[22]. Direct laser photoexcitation of carriers in the QCL also allows modulation of the free-carrier absorption and hence THz power[23], although the bandwidth is limited by carrier recombination (~700 ns) and thermal (~µs) effects[24].

In this article, we demonstrate a previously unexplored approach to modulation of a QCL, through ultrafast acoustic modulation of the electronic states and carrier transport. Acoustoelectric effects are important in microwave technologies, for example in filters[25], delay lines[26] and modulators[27]. It has been demonstrated previously that bulk acoustic waves with ~100-GHz frequencies can dynamically alter the electrical and mechanical properties of semiconductor heterostructures on length-scales comparable to the epitaxial-layer thickness, which can modulate the electron transport in tunnelling[28] and planar[29] devices, as well as the light emission from quantum dot lasers[30]. Here, we use an optically generated picosecond acoustic (strain) pulse[31], propagating along the growth direction of a QCL, to modulate its band structure and electron transport on ultrafast timescales. This enables ultrafast THz-power modulation without the bandwidth limit imposed by parasitic electronic effects. We discuss the results in terms of a theoretical model that considers the strain-induced perturbation of electronic states via the deformation-potential electron–phonon interaction.

## Results

**Acoustic pulse-induced modulation.** Figure 1a shows schematically the experimental arrangement used to launch acoustic strain pulses into a QCL and to observe dynamically their effect on its optical and electronic characteristics. An aluminium-film acoustic transducer was deposited on the bottom surface of the semi-insulating GaAs substrate of a 2.5–2.75 THz QCL containing 88 periods of the active-region heterostructure[32], as shown in Fig. 1b. The THz QCL device with dimensions of $(2000 \times 150)$ µm² was mounted onto the cold finger of a cryostat, with a $1.0 \times 0.3$-mm² slit aperture allowing optical access to the aluminium transducer layer. Longitudinal-acoustic pulses with a single-cycle (bipolar) form and ~20-ps[33] duration were generated in the device by optically exciting the transducer using ultrafast infrared pulses from an amplified Ti:Sapphire laser (see Section "Methods"). The optical pulses were focused using cylindrical lenses to form a beam with dimensions of ~1.1 mm × 0.3 mm at the QCL position, resulting in ~50% spatial overlap between the optical beam and the QCL device. Perturbations to the electron transport in the QCL were monitored via the device voltage, with the THz emission being simultaneously monitored on a Schottky-diode detector. The $L–I–V$ characteristics and the band structure of the QCL device are shown in Fig. 1c, d, respectively.

Figure 2a shows the time-varying THz emission, $L(t)$, and the QCL voltage, $V(t)$, recorded with an optical pulse energy of 10 µJ incident on the Al-film transducer and with a quiescent QCL bias of $V_b = 5.58$ V ($V_3$ in Fig. 1c). The first pulse in both $L(t)$ and $V(t)$ appears 32 ns after the initial optical excitation (at $t = 0$ ns, not shown), corresponding to the transit time of the acoustic wave, propagating at the speed of longitudinal sound in GaAs ($4800$ ms$^{-1}$), through the ~150-µm-thick substrate. Subsequent echo-pulses occur at 70-ns intervals, owing to acoustic reflections between the top metallic (Ti/Au) contact of the QCL and the bottom of the substrate. We assume that the acoustic pulse is not significantly perturbed as it propagates through the QCL structure. This is consistent with pump–probe measurements of the acoustic phonon modes of typical QCL structures[34] which show phonon lifetimes less than or equal to the time for an acoustic wave to propagate once through the structure, and indicates narrow phonon stop bands in the frequency range of interest. As the acoustic wave propagates through the QCL, it perturbs the electrical transport and modulates the THz emission. Although the acoustic pulse has a duration of ~20 ps, the duration of the acoustically induced effect is determined by the round-trip transit time of the acoustic wave through the 13.9-µm-thick QCL active region, which was calculated to be ~6 ns using the speed of sound in GaAs. This is shown experimentally in Fig. 2b. For these operating conditions, with the QCL bias exceeding that required for subband alignment $V_A$, the power modulation $\Delta L$ is negative in sign.

The perturbation of the electron transport also results in a voltage modulation, $\Delta V(t)$, as shown in Fig. 2a where a positive modulation ($\Delta V > 0$) is observed. This implies an increase in resistance (assuming constant current) caused by the acoustic wave passing through the heterostructure; this observation is consistent with predictions from a time-dependent perturbation model of the QCL under perturbation by a strain pulse (see Section "Theoretical analysis on the origin of voltage perturbations"). The fast THz power modulation is followed by a broader positive peak after the acoustically induced pulse in $V(t)$ has ended, and coincides with the small negative ringing in $V(t)$ that we attribute to resonance in the electrical circuit.

Closer inspection of the voltage pulses (Fig. 2b) reveals they comprise an initial ~3-ns pulse, followed by a ~3-ns-long shoulder owing to reflection of the strain pulse in the opposite direction. The reflected pulse amplitude is ~40% that of the forward pulse, which is consistent with the amplitude reflection coefficient for the strain pulse at the GaAs/Au interface,

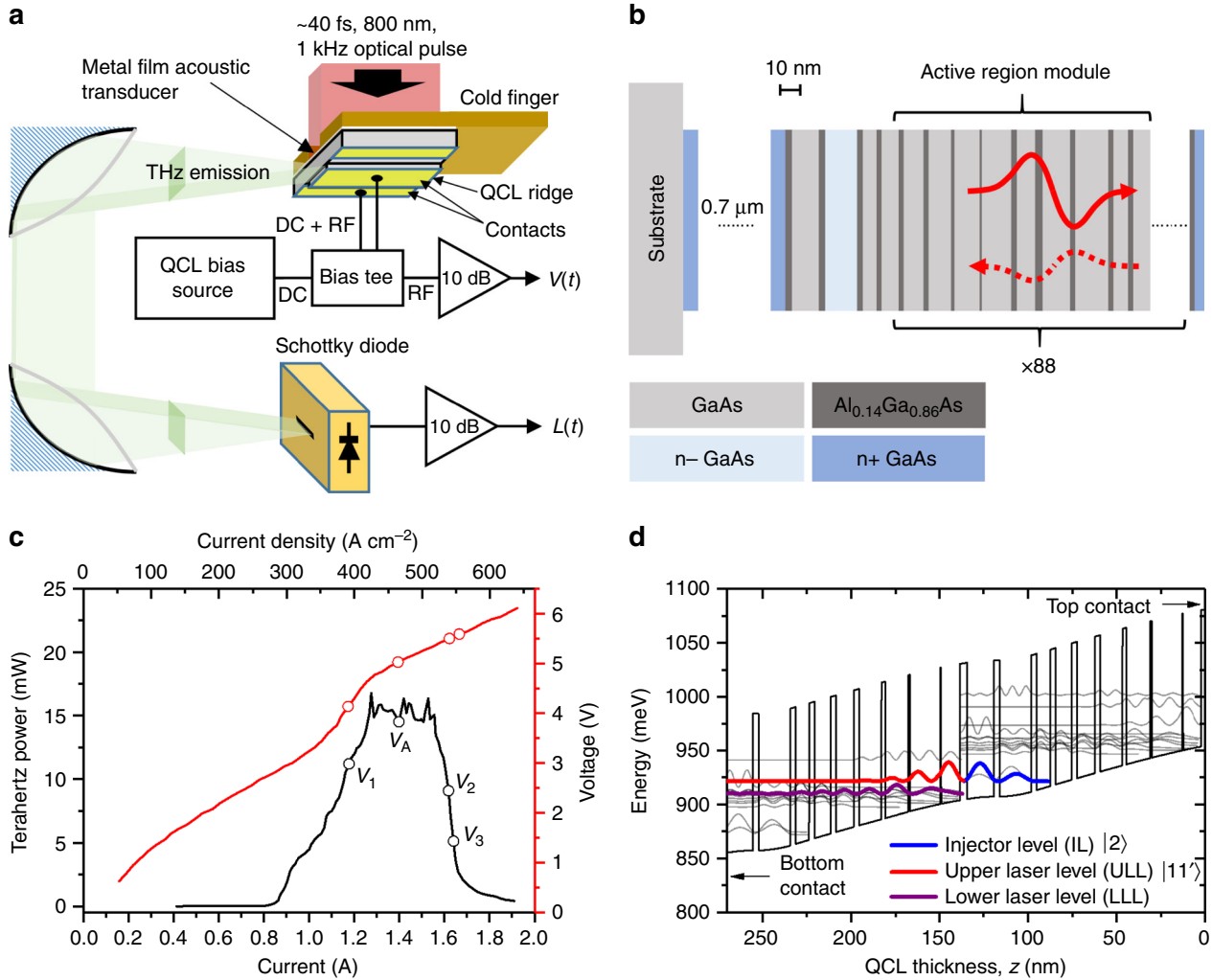

**Fig. 1 Experimental arrangement and QCL device structure and characteristics. a** Experimental arrangement for measuring the optical and electronic perturbation of a THz QCL by laser-generated picosecond acoustic pulses. The sample is mounted on the cold finger of an optical cryostat at an operating temperature of 15 K. **b** Schematic diagram of the QCL device structure showing the transmitted (solid red line) and reflected (dashed red line) strain pulses. **c** L–I–V characteristics of the QCL in the absence of acoustic pulses, measured at a temperature of ~10 K and driven with 50-μs pulses at a 1-kHz repetition rate. Labels correspond to the different QCL biasing conditions used in experiments: ($V_1 = 4.13$ V, $V_A = 5.02$ V, $V_2 = 5.49$ V and $V_3 = 5.58$ V). **d** Simplified QCL band structure showing two periods of the structure with labelled injection level (IL-$|2\rangle$), upper laser level (ULL-$|11'\rangle$) and lower lasing level (LLL). Source data for **c** and **d** are provided in ref. [50].

$r = (Z_{Au} – Z_{GaAs})/(Z_{Au} + Z_{GaAs})$, where $Z_{Au} = 63 \times 10^6$ kg m² s⁻¹ and $Z_{GaAs} = 25 \times 10^6$ kg m²s⁻¹ are the acoustic impedances of Au and GaAs, respectively.

It is important to note that, owing to the short (~ps) carrier lifetimes in QCLs, a strain-induced band structure deformation will perturb the carrier transport on ultrafast timescales. The rise time of the THz power modulation in our device will therefore be limited fundamentally by the acoustic-pulse duration (~20 ps) and its transit time through an individual injection region of the heterostructure stack (see Section "Theoretical analysis"). Nevertheless, the perturbation in $V(t)$, shown in Fig. 2a, b, exhibits a rise time that is limited to ~800 ps by parasitic device impedance (see Section "Methods"). The rise time of $L(t)$ is similarly limited by the Schottky-detector response, rather than the timescales of the underlying acoustoelectric processes.

Figure 3 shows the effect of the initial acoustic pulse on $L(t)$ and $V(t)$ (3a and 3b, respectively), recorded with the QCL bias, $V_b$, set either below or above the subband alignment voltage, $V_A$. The acoustic pulse induces an increase in $V(t)$ in all cases,

whereas the sign of $\Delta L$ depends on the QCL bias: below subband alignment, $\Delta L > 0$, and above subband alignment, $\Delta L < 0$. Furthermore, for measurements performed with $V_b = V_A$ (not shown) the THz-power modulation was indistinguishable from noise ($\Delta L \approx 0$). These observations can be reconciled using a quantitative phenomenological analysis based on the measured relationship between the bias voltage and the unperturbed THz power (see Section "Relation between voltage and THz power modulation"). It is important to note that both the voltage and power modulations are unipolar in nature, although distortion by ringing due to resonance in the electrical circuit does cause the modulation in $L(t)$ to appear bipolar at certain biasing conditions. However, as is evident in Fig. 3a, b (in which the shaded areas indicate the times at which the acoustic pulse will be acting on the QCL active region), it can be seen that these ringing effects occur after the acoustic pulse has left the QCL ridge. As such, these effects are attributed to the active region relaxing to its unperturbed state, and not attributed directly to the modulation due to the passage of the acoustic pulse through the active region.

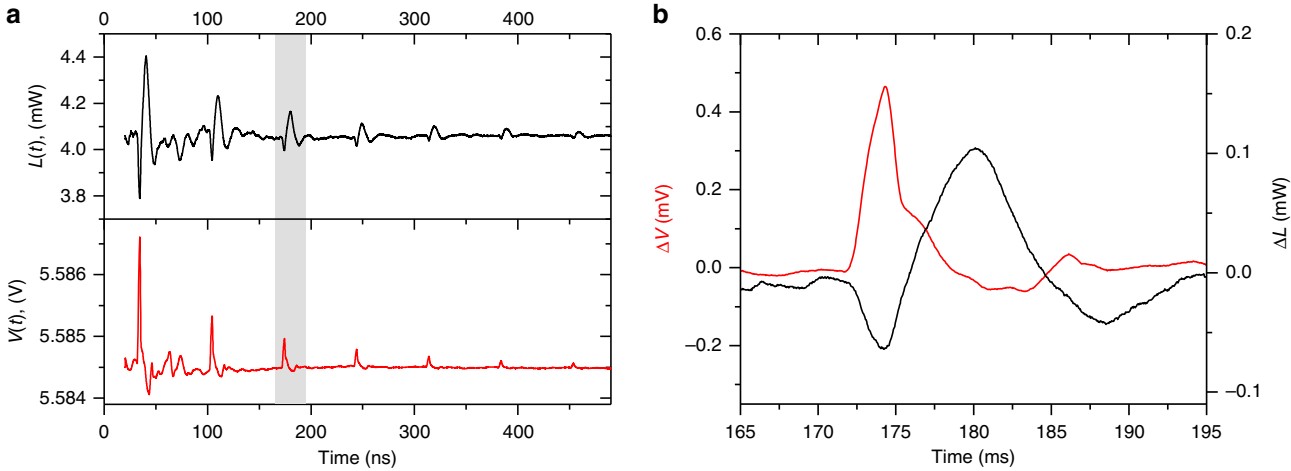

**Fig. 2 Observation and measurement of the acoustic modulation on the QCL device. a** Temporal response of the THz emission, $L(t)$, (black trace, top) and QCL voltage, $V(t)$, (red trace, bottom) to incident strain pulses generated with an optical pulse energy of 10 μJ at a QCL bias $V_b = V_3$. The initial strain pulse, arriving at time $t = 32$ ns, is generated by laser impact on the Al-film transducer at $t = 0$ and the following pulses, occurring every 70 ns, are due to multiple reflections in the device. **b** Shows the change in the $L(t)$ and $V(t)$ responses ($\Delta L$ and $\Delta V$, respectively) to an individual acoustic pulse for the region highlighted in (**a**). Data have been smoothed using a 20-point rolling average filter. Source data are provided as a Source Data file.

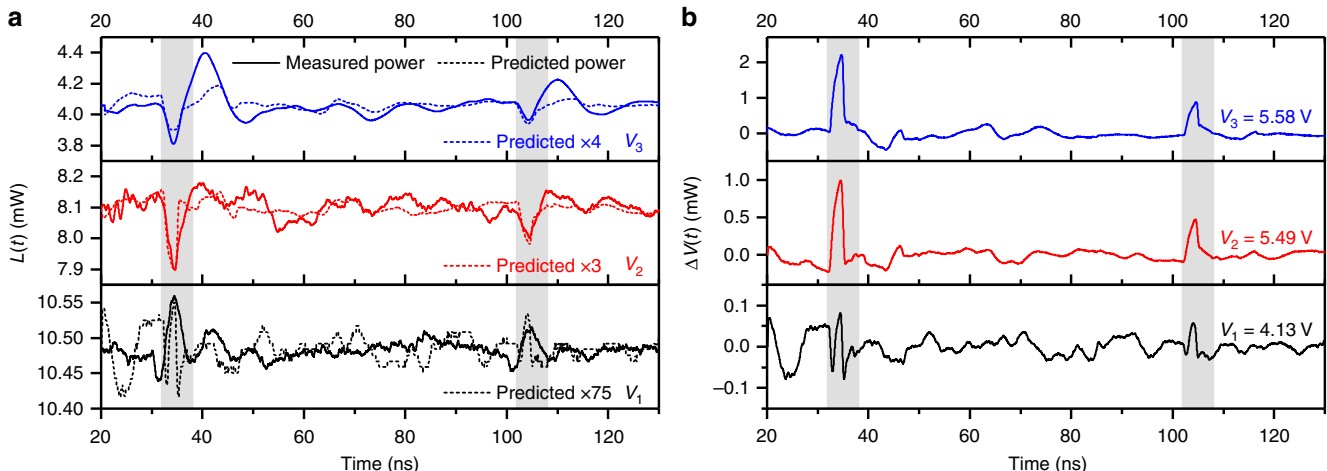

**Fig. 3 Measurement and prediction of effect of the acoustic modulation.** Temporal responses of the QCL to the first two strain pulses at three different QCL operating biases $V_b = V_1$ to $V_3$: **a** THz power modulation, $L(t)$ (solid lines) and **b** QCL voltage perturbation, $\Delta V(t)$. Predicted power modulation waveforms are shown in (**a**) (dashed lines), with amplitudes normalised to the experimental data by factors of 75, 3 and 4 for $V_1$, $V_2$ and $V_3$, respectively. Highlighted regions indicate the temporal position of the acoustic modulations. Incident strain pulses were generated with an optical pulse energy of 10 μJ. Data have been smoothed using a 20-point rolling average filter. Source data are provided as a Source Data file.

Figure 4 shows the amplitude of $\Delta L$ and $\Delta V$ as a function of the optical pulse energy measured outside the cryostat. Since the amplitude of the strain pulse is proportional to the laser fluence incident on the Al-film transducer[31], we conclude that both perturbations increase linearly with strain amplitude. No temporal shift relative to the optical excitation pulse was observed in either the $\Delta V$ or $\Delta L$ responses in any of these measurements, indicating that the strain amplitude was insufficient to cause nonlinear acoustic propagation or shock wave formation. The maximum THz power modulation in Fig. 3a, $\Delta L = -0.25$ mW, was observed for QCL bias $V_3$, where the unperturbed THz power was $L \approx 4$ mW, giving a fractional change $|\Delta L/L| \approx 6\%$. This modulation depth was measured using a 10-μJ optical pulse, although it was found that the optical pulse energy could be increased to at least 24 μJ without damage to the Al transducer layer. As there is no evidence of nonlinear acoustic effects, it is reasonable to assume, via extrapolation, that >15% THz modulation depth could be achievable.

**Theoretical analysis on the origin of voltage perturbations**. QCLs are resonant-tunnelling devices, in which quantised electron states localised in adjacent periods are brought into resonance at the alignment bias, $V_A$, as shown in Fig. 1d. Inter-period transport is dominated by resonant tunnelling across an injection barrier, and models of the transport and band structure typically exploit the structural periodicity of the device[35]. However, the acoustic-strain pulses induce a localised perturbation, which causes electric-field-domain formation[36] and breaks this periodicity. The ~20-ps acoustic pulse duration is comparable to the state lifetimes within the QCL, and therefore a time-dependent perturbation (TDP) model[37] is needed to calculate the transition probabilities for resonant tunnelling between two periods of the QCL.

The effect of the acoustic pulses on the inter-period tunnelling rate between an injector state (IL) in the first period, and the upper laser level (ULL) in the second period, denoted $|2\rangle$ and $|11'\rangle$, respectively, was calculated using a TDP approximation

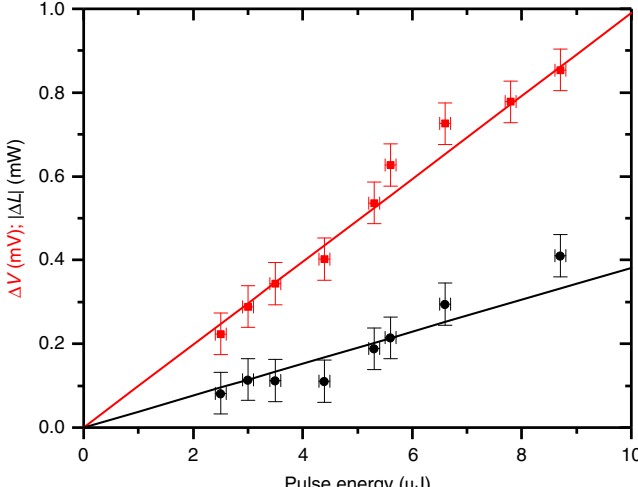

**Fig. 4 Amplitude of acoustic modulation measured as a function of optical pulse energy.** Amplitude of voltage $\Delta V$ (red squares) and power $\Delta L$ (black circles) perturbation signals for the first strain-induced pulse, measured at a bias voltage $V_b = 5.38$ V. Error bars represent the measurement errors on the pulse energy and amplitudes. The lines are linear fits to the data. Source data are provided as a Source Data file.

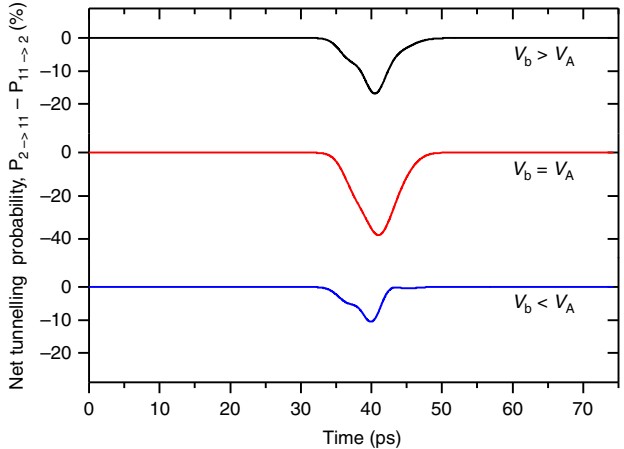

**Fig. 5 Net tunnelling probabilities for the perturbed QCL device using a time-dependent model.** Time-dependent net tunnelling probabilities between the injection level ($|2\rangle$) and the upper laser level ($|11'\rangle$) due to transit of the acoustic wave, calculated using a time-dependent perturbation model for QCL biases below subband alignment ($V_b < V_A$, $V_b = 3.53$ kV cm$^{-1}$), at alignment ($V_b = V_A$, $V_b = 3.63$ kV cm$^{-1}$) and above alignment ($V_b > V_A$, $V_b = 3.85$ kV cm$^{-1}$). The timescale of the simulation is set so that the pulse can propagate over two adjacent periods. Source data are provided as a Source Data file.

(see Section "Methods"). The amplitude of the perturbation caused by the acoustic wave is typically ~0.01–1 meV[38], corresponding to strain amplitudes in the range $10^{-6}$–$10^{-4}$ (using the deformation potential of GaAs is ~10 eV per unit strain), which is consistent with piezospectroscopic measurements using metal transducers on GaAs[38]. Although the amplitude of the perturbation may be comparable with the anti-crossing energy (~0.4 meV), our TDP model gives a simpler and more intuitive analytical approach than other methods such as Landau–Zener theory[39,40] or a full time-dependent Schrödinger solution.

Figure 5 shows the calculated time-dependent net tunnelling probabilities between states $|2\rangle$ and $|11'\rangle$ arising due to propagation of the acoustic wave through the heterostructure, under the assumption of equal initial state populations. Device operation was considered at three bias points: below the alignment bias, $V_b < V_A$; at alignment, $V_b = V_A$ and above alignment, $V_b > V_A$. In all three cases the duration of the simulated perturbation is commensurate with the ~20 ps acoustic pulse duration and causes a greater increase in reverse tunnelling than in forward tunnelling, leading to a net reduction in forward-tunnelling probability. This is consistent with the experimentally observed increase in device voltage $V(t)$ above the laser threshold for a constant driving current. The time-dependent net tunnelling probabilities between the injector and upper lasing states were found to be perturbed by up to ~40% by the presence of the acoustic wave; for comparison, the same calculation between the upper and lower lasing states yielded a ~0.001% effect.

It is important to note that the model considers carrier injection into the ULL only and not into parasitic current pathways (i.e., states not involved in photon emission). Experimentally, the net current injection is constant, as the perturbed region has a much lower impedance than the source and the rest of the device. The corresponding voltage change across the perturbed region modulates the state populations and hence photon generation in the device. This implies a qualitative explanation for the observed polarity of the THz-power modulation: below $V_A$, an increase in voltage causes an increase in THz power, whereas the converse applies above $V_A$.

**Relation between voltage and THz power modulation.** The interaction between acoustic waves and the QCL band structure at the quantum level is complex, with multiple subbands perturbed by the propagating strain wave. A direct quantum analysis of the THz power perturbation is extremely difficult. However, a quantitative phenomenological analysis can be obtained from the measured relationship between the bias voltage, $V_b$, and the unperturbed THz power, $L$ (derived from Fig. 1c), as this encapsulates the underlying quantum phenomena within the single observable parameter, $L$. A linear perturbation approximation then allows a direct prediction of the change in THz power as the voltage varies dynamically through a small perturbation around the bias point.

Owing to the different spatial localisations of the static voltage ($L$–$V$) measurements and the temporal voltage responses $V(t)$, it was necessary to equate them by converting the voltage to the internal electric field for both data sets. For the unperturbed $L$–$V$ data measurements of Fig. 1c, the voltage is dropped across the QCL ridge of height $D = 13.9$ μm, assuming negligible contact resistance. This gives an internal field of $F_0 = V_b/D$ throughout the device, assuming a single electric-field domain. For the temporal responses in Fig. 3b, the voltage perturbation is assumed to correspond to the average internal field perturbation, localised over the spatial extent of the acoustic wave, $d = 96$ nm (determined from the pulse duration and the speed of sound in GaAs). The spatially averaged local field perturbation is then $\Delta F(t) = \Delta V(t)/d$.

In this way, the measured voltage perturbation across the QCL can be linked directly to a perturbation in THz emission power according to the relationship $\Delta L(t) = (dL/dF_0)\Delta F(t)$, where $dL/dF_0$ is the slope efficiency per period of the active region heterostructure.

The predicted power modulation resulting from this analysis is shown in Fig. 3a (dashed lines) for the three biasing conditions, $V_b = V_1$ to $V_3$, along with the corresponding experimental measurements (solid lines). The predicted signal shape and

polarity match well with the measurements at bias points $V_2$ and $V_3$, for both the original acoustic pulse and the subsequent echoes. At these biases, the slope efficiency per period is large; $dL/dF_0 \approx -0.98$ mW per kV cm$^{-1}$ and $dL/dF_0 \approx -0.21$ mW per kV cm$^{-1}$ for $V_2$ and $V_3$, respectively. As such, the voltage perturbations ($\Delta V \sim 1.2$ mV and $\sim 2.2$ mV, respectively) are predicted to induce THz-power modulations of $\Delta L \approx -120$ μW and $\Delta L \approx -50$ μW, respectively. These predictions are, respectively approximately two and five times smaller than the experimental values. It is hypothesised that this discrepancy arises from the assumption that the acoustic pulse affects the emission sequentially from only a region of length $d$, comparable with one period of the QCL heterostructure. Indeed, owing to the nature of electron transport in a QCL, it is possible that periods downstream of the acoustic pulse will also be affected. We also do not consider the possibility of electric-field-domain formation due to the passage of the acoustic pulse through the QCL, which would disrupt carrier transport in a manner not predicted by this model and would change the effective localised quiescent field. At bias $V_1$ (below band alignment), the predicted power modulation is small and positive, which is also borne out by our measurements. At lower biases, the modulated signal is affected more by noise in the experimental system, making it more difficult to determine the effect of the acoustic pulse on $L(t)$. Nevertheless, it was observed that the modulation of power is consistently positive in the time window where the acoustic modulation occurs. This small power predicted modulation is consistent with the small acoustically induced voltage modulation ($\Delta V \le 0.1$ mV; black line, Fig. 3a), as well as the small slope efficiency of the device at this bias, $dL/dF_0 \approx 0.08$ mW per kV cm$^{-1}$.

## Discussion

We have reported the modulation of a QCL using optically generated bulk acoustic-phonon pulses propagating through the QCL heterostructure. This has been used to alter the electronic states dynamically on ultrafast timescales, thereby modulating the THz emitted power by up to 6% of the unperturbed value. A time-dependent model of the strain-induced band structure perturbation has revealed how the acoustic deformation potential leads to a net transient increase in device resistance when operated above threshold. This has been observed experimentally as a time-dependent THz-power modulation, occurring alongside a transient increase in device voltage under constant driving current.

In the experiments reported here, the rise time of the QCL-voltage perturbation was limited to ~800 ps by the parasitic device impedance, whereas the observable rise time of the THz modulation was limited by the Schottky-detector response. However, it is important to note that the electron dynamics and the deformation-potential coupling processes in QCLs are inherently extremely fast (~ps timescales). The theoretical maximum modulation speed will therefore be determined by the transit time of acoustic pulses through the layers in the QCL heterostructure that are active in the modulation process. For a typical ~20-nm-thick layer this transit time is ~4 ps, whereas in our experiment the round-trip time through the 13.9-μm-thick QCL stack was ~6 ns. In principle, this could be reduced substantially by use of a QCL with a thinner active region[41] or by engineering a device in which only a few periods or heterostructure layers are sufficiently responsive to the acoustic perturbation. Combined with shorter acoustic pulses, generated using a thinner transducer film or through direct UV excitation of the substrate[42], this opens up the possibility of modulation frequencies exceeding ~200 GHz.

The authors note that in experiments demonstrating direct acoustic-to-THz conversion the effect is weak compared to emission from a QCL[43], and that even the 6% modulation demonstrated here is considerably greater than the total emission from these direct acoustic-to-THz conversions. It is speculated that through quantum engineering of the QCL heterostructure it should be feasible to achieve even larger modulation depths than those observed here. One possible approach would be through modification of the injector levels. By using thinner injection barriers (such as ~24–30 Å barriers theoretically proposed for resonant phonon depopulation structures[44], or the ~44 Å barriers used for high performance QCL devices based on scattering-injection structures[45]), it is hypothesised that the induced deformation potential could result in greater perturbation of electron transport through the structure. Another possibility to increase both the amplitude and frequency of the modulation would be to set up an acoustic standing wave in the QCL, with antinodes at the positions of the injection barriers. In this case, the modulation would occur at the acoustic wave frequency, up to 100 s of GHz, and the amplitude would be increased compared to our measurement here because all of the periods of the QCL would be modulated by the acoustic wave simultaneously. In practice this could be accomplished, for example, using acoustic Bragg mirrors grown above and below the QCL active region stack; these Bragg mirrors could also be highly doped, allowing them to act as a THz plasmonic waveguide. The frequency of the acoustic standing wave is determined by the period of the Bragg mirrors, with a 100 GHz standing wave requiring each period of the Bragg superlattice to be ~25 nm thick.

Our alternative approach to fast modulation also opens up the possibility of direct frequency modulation of THz QCLs through acoustic perturbation of the electronic states responsible for optical gain. These techniques could find potential applications in high-speed communications and high-resolution spectroscopy. For such practical applications, the laser pumped acoustic transducer used in our experiments could be replaced by an on-chip acoustic source, for example a piezoelectric bulk acoustic wave transducer or a semiconductor phonon laser (SASER) device[46]. For example, an AlN thin-film transducer with a few volts applied could be used to generate strain amplitudes up to 10$^{-4}$, comparable to the strains used in our experiment and theory. Piezoelectric transducers have been demonstrated to operate up to ~96 GHz[47], while SASER devices have been demonstrated to operate up to 325 GHz[46], implying that using either of these methods as the source of the modulating strain-pulse could enable larger (although still comparable) modulation bandwidths than that achieved here.

## Methods

**QCL fabrication and optical excitation**. The QCL structure used in this work was based on a 9-well GaAs/AlGaAs active region emitting at a frequency 2.5–2.75 THz[32]. The active region layer sequence is 10.6/**0.5**/17/**1**/13.5/**2.1**/12.4/**3.1**/10/ **3.1**/9/**3.1**/7.5/ **3.1**/17.8/**3.1**/15.2/**4.1** nm, with Al$_{0.14}$Ga$_{0.86}$As barriers indicated in bold and the uniformly Si-doped layer ($N_d = 3.2 \times 10^{16}$ cm$^{-3}$) underlined. A 13.9-μm-thick heterostructure was grown using molecular-beam epitaxy on a semi-insulating GaAs substrate, and processed into a surface-plasmon ridge-waveguide structure with dimensions (2000 × 150) μm$^2$. The substrate was thinned to ~150 μm to aid thermal dissipation, and a 100-nm-thick Al film acoustic transducer was deposited on the polished back surface. The QCL was mounted within a Janis ST-100 liquid-helium cryostat, with a slit aperture in the cold finger permitting optical access to the surface of the metal film transducer directly underneath the QCL ridge. Acoustic pulses were generated in the transducer using ~40-fs pulses from an amplified Ti:Sapphire laser, with wavelength centred at 800 nm and a pulse-repetition rate of 1 kHz. Cylindrical lenses were used to focus the laser beam to an elliptical spot on the Al film measuring ~1.1 mm × 0.3 mm. Optical pulse energies of up to ~10 μJ were used, corresponding to a maximum fluence of $F = 3$ mJ cm$^{-2}$ on the transducer, obtained using $F = P(1 - R)/A$, where $P$ is the average pump laser power, $R$ is the total reflectance due to the focusing lens and cryostat window and $A$ is the area of the laser spot on the Al film.

The QCL was operated at a heat-sink temperature of 15 K and driven with 50-μs pulses synchronised to the 1-kHz repetition rate of the optical excitation pulses. The ac-component of the QCL terminal voltage was separated from the

quasi-dc bias using a microwave bias tee with a bandwidth of 14 GHz. These high-frequency contributions induced by the acoustic wave were amplified with a low-noise voltage amplifier and measured using a 12.5-GHz digital-sampling oscilloscope. The reactance of our packaged QCL device is dominated by the parasitic inductance of the wire bonds, which is calculated to be $L_w$~1 nH[48]. At the operating bias (for example $V_b = V_3$) the differential resistance of the device is measured to be $R_d = 2.1$ Ω (see Fig. 1c), giving a time constant for the voltage signal $L_w/R_d$ ~500 ps. A pair of off-axis parabolic mirrors were used to collimate and focus the THz radiation emitted by the QCL into the WR-0.4 diagonal feedhorn of a fundamental 1.8–2.8 THz Virginia Diodes Schottky diode detector. The high-frequency transient response of the Schottky detector was amplified using voltage amplifiers with a bandwidth of 14 GHz and monitored using the sampling oscilloscope.

**Time-dependent perturbation theory**. A one-band effective-mass Schrödinger–Poisson model was used initially to calculate the wave-function basis for the unperturbed QCL, and tight-binding boundary conditions[49] were used to confine electrons within a single period of the device. The effect of the acoustic pulse on transport between the electronic states was calculated as a function of time by taking the first two correction terms in a TDP expansion[37]. In these calculations the time profile of the acoustic-strain pulse was modelled as the derivative of a Gaussian function of total width 20 ps (shown in Fig. 1b), which is a satisfactory approximation to the actual shape. The acoustic pulse induces a localised perturbation of amplitude $\eta \Xi_D$, where $\eta$ is the strain amplitude and $\Xi_D$ is the deformation potential (~10 eV per unit strain in GaAs). The resulting TDP transition probabilities are denoted $P_{2\rightarrow11'}$ and $P_{11'\rightarrow2}$ for tunnelling in the forward and reverse directions across the injection barrier, respectively. The change in current density then takes the functional form $J \sim n_2 P_{2\rightarrow11'} - n_{11'} P_{11'\rightarrow2}$, where $n_2$ and $n_{11'}$ are the populations of the states before and after the barrier, respectively. These populations are highly dependent on the applied bias and can only be determined using a detailed carrier-transport analysis. However, the net tunnelling perturbation can be estimated by assuming an identical population in each subband.

## Data availability

The data that support the plots within this paper and other findings from this paper are available at https://doi.org/10.5518/579 (ref. [50]).

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

## Acknowledgements

The authors thank N. Hunter for his previous work on this programme, and useful discussions. Financial support was received from the UK Engineering and Physical Sciences Research Council (grant references EP/M016161/1, EP/M01598X/1, EP/P001394/1 and EP/P021859/1), UK Research and Innovation (Future Leaders Fellowship MR/S016929/1), the UK Space Agency (contract NSTP3-FT2-002), the Royal Society (Wolfson Research Merit Award WM150029) and from Leeds International Research Scholarships (LIRS).

## Author contributions

A.K., J.C. and P.D. conceived the idea and designed the experiment. A. Dunn, P.D., C.P. A.K. and A.V. developed the experimental set-up and performed the measurements. Data were analysed by A. Dunn and C.P. with support from A.K., P.D., A.A. and J.C. The QCL structure was grown by L.L. under the supervision of E.H.L. Devices were processed by M.S., I.K. and A. Dunn under the supervision of E.H.L and A.G.D. Theoretical simulations were performed by A. Demić and A. Dunn, with support from D.I. and A.V. The paper was written by A. Dunn, P.D., A.V., A. Demić and A.K. with contributions from all authors.

## Competing interests

The authors declare no competing interests.
