## [Peer Review File · Nature Communications]

Reviewers' comments:

Reviewer #1 (Remarks to the Author):

The manuscript by Dunn et al presents a new method of modulating the output of a THz quantum cascade laser. The authors report the use of acoustic waves to perturb and modulate the output power of a THz QCL by up to 5%. The authors propose that the modulation is the results of strain associated with the acoustic waves perturbing the electronic band structure. Their assertion is backed up by time dependent perturbation theory calculations. This is a novel new mechanism of QCL modulation, and both the experimental and theoretical work are thorough and well presented.

I feel that this manuscript is appropriate for publication in Nat Comms subject to a couple of suggested minor corrections:

1. The use of average power when describing the optical pump is not really that meaningful, given the pulsed nature of the experiment and size of the device. It would be better to use fluence or energy instead of average power through the text. The average power and spot size could still be given in the Methods section, where the fluence is currently stated.
2. In the discussion section the authors comment briefly on approaches that could be made to improve the modulation depth, and approaches to making an electrically controlled device (i.e. using piezoelectric transducers instead of the bulky amplified laser). These approaches are clearly very exciting for communications applications. Can the authors comment on the cost to modulation bandwidth on using these alternative approaches?

Reviewer #2 (Remarks to the Author):

The Authors present an interesting and original study where optically-generated acoustic phonon pulses are used to affect the QCL bandstructure. Fast amplitude modulation of the laser signal is observed . The experiments and the results are surely worth publication, but there are a number of points throughout the paper that in my opinion need clarification.

- 1 The experimental setup lacks some details, especially regarding the coupling of the near-ir ultrafast pulses to the back of the QCL. How large is the spot? is the whole laser modulated? if not, what is the estimated overlap ? The informations are in the Methods section, I think they should be present in the setup figure 1.
- 2 FIG 1C , the LIV plot looks a bit weird, like the graphs are shifted with respect to the axes because the I-V curve seems not to go trough zero at zero bias. Please explain/correct
- 3 Fig.4 The linear fit to the data seems not to be the correct choice. There are clear oscillations both in voltage and power that produce a "steplike" behaviour. Any more comments on the origin of these steps? The assumption of linearity is maybe not fitting in the present case?
- 4 The speculations about the possibility to modify the QCL band structure to introduce , for example, acoustic Bragg mirrors are in my opinion a bit naive, QCL structure is extremely delicate and barrier thinning and or other modifications would have a sizeable impact on the performance of the device. Such speculations, if present in the paper, should be supported by a little bit more quantitative figures, like by how much barriers would be thinned and teh period of the acoustic Bragg mirror.
- 5 Regarding the simulations that include the acoustic wave perturbation to the QCL potential, which value has been used for eta, the strain amplitude? is the value used compatible with what produced by the laser pulses? would there be a way to assess more quantitatively the strain amplitude and characterize better the acoustic wave perturbation which impinges onto the QCL? Maybe looking at some Raman measurements could help.
- 6 Still concerning the quantitative amplitude of the strain, this would be useful also to support the claims of a possible use of this effect employing on-chip solutions like piezo transducers.

7 In the references , I think there could be space for the paper by Bruchhausen et al., Journal of Applied Physics 112, 033517 (2012), who investigated coherent acoustic phonons in THz qcls by means of pump probe spectroscopy.

To conclude, I think the paper needs more quantitative assessments on the simulations and clarifications on the previous points. The claim of applicability of these technique to modulate actual devices are in my opinion excessive and they should be turned a bit down. the bottom access of this kind of technique or others presents some serious troubles in terms of efficient heat sinking of the QCL.

The paper cannot be published in the present form.

Reviewer #3 (Remarks to the Author):

Report on the manuscript entitled « High-Speed Modulation of a Terahertz Quantum Cascade Laser by Coherent Acoustic Phonon Pulses » by Dunn et al.

The paper reports on the modulation with acoustic waves of the THz emission of a quantum cascade laser (QCL). This modulation is achieved at the MHz-sub-GHz frequency. The maximum modulation reached is around 5%. The acoustic waves (generated by a femtosecond laser) are generated in a thin metallic film mechanically coupled to the QCL. Once produced by the femtosecond laser, these acoustic waves enter then in the QCL and modulate the electronic structure through the deformation potential mechanism. The authors indicate that the tunneling between the injection levels and the upper QCL level is the mainly affected physical parameter. On the basis of this deformation potential coupling, the authors propose a semi-quantitative model (based on computed time-dependent Schrödinger equations) describing the modulation of the THz emission.

I don't know if it is relevant from the QCL applications point of view, but the physics is potentially interesting and following the previous work of the authors Ref 28-29, interesting physics of electron-phonon in semiconductor hetero-structures is discussed. I have some comments regarding the results analysis and the discussion and some issues need to be addressed before publication:

1) The authors claim they demonstrate a new mechanism. I would say that at least two references probably forgotten by the authors already reported acousto-electric effects or transient built-in field leading to the modulated/transient THz emission. The main difference in the submitted paper comes for the system in which such acousto-electric effect takes place. In the references cited below, such acousto-electric effect was demonstrated in semiconductors hetero-structures (Armstrong et al). In Van Capel et al's paper, semiconductor superlattices were studied ; the acousto-electric effect is not directly discussed but the physics is very close even if in the latter one, piezoelectric effect comes in addition.

References :

Armstrong et al, Observation of terahertz radiation coherently generated by acoustic waves, NATURE PHYSICS VOL 5 285-288 (2009)

Van Capel et al, Correlated terahertz acoustic and electromagnetic emission in dynamically screened, InGaN/GaN quantum wells PHYSICAL REVIEW B 84, 085317 (2011)

I strongly encourage the author to cite/comment this work to make the survey more complete. Moreover, they also should then demonstrate more clearly the new opportunities that their experiments provide for the THz modulation point of view.

2) I do not understand the advantage of using acoustic waves when the THz modulation lasts around 6 ns (166 MHz), i.e. an order of magnitude slower than what can be achieved with electrical modulation (Ref 19). Could the authors comment on this point ?

3) Regarding the application, why did the authors not study a mid-infrared QCL working at room temperature to investigate this acoustoelectric effect ?

4) It is not clearly explained how a 20 ps acoustic pulse becomes a bipolar 6 ns pulse in the structure. On page 5, the authors say that the "bipolar" shape of the THz "pulse" (fig 2) is due to the existence of an acoustic reflection. The L curve (black curve) clearly exhibits a change of sign which is not the case of V. This part is not convincing. Maybe the time window of the inset of Fig 2 should be increased to evidence more clearly the change of sign ?

5) The authors say that increasing V_3 gives rise to a decrease of L and V_1 the converse effect. I see the first effect but not the converse one. Why ?

The authors state the acoustic pulse is bipolar, but due to electronic diffusion, this should not be the case (see Ref 40). So, could this asymmetry be responsible for an asymmetry in the phonon-induced tunneling (reverse/forward) in the sense that the compressive and expansion strain tails do not have the same spatial extension and then the deformation-potential perturbation does not have the same spectrum (i.e. not the same frequency components) ? This should lead to a different rate of transition in the time-dependent Schrödinger equation.

6) The authors calculate the acoustically-induced tunneling (injection level towards the upper level). There is no discussion regarding the perturbation of the dipolar matrix element involved in the radiative recombination ? Why ?

Response from Authors to Reviewers' Comments

We would like to thank the reviewers for their supportive comments on our manuscript. We are pleased they judge the work to be '*appropriate for publication in Nature Comms*' (reviewer 1) and '*surely worth publication*' (reviewer 2), while reviewer 3 notes '*some issues need to be addressed before publication*'. These issues, and others raised by the reviewers are addressed below. (Reviewers' comments are given in italics throughout.)

Reviewer #1

We would like to thank the reviewer for their very supportive comments.

1. *The use of average power when describing the optical pump is not really that meaningful, given the pulsed nature of the experiment and size of the device. It would be better to use fluence or energy instead of average power through the text. The average power and spot size could still be given in the Methods section, where the fluence is currently stated.*

The authors agree that quoting the average power is not an ideal measure for a pulsed laser source. Therefore in the revised manuscript we now quote the pulse energy (instead of average power) of the optical laser measured outside of the cryostat. The x-axis values on Figure 4 have been changed to pulse energy accordingly. Details are provided in the Methods section regarding the pulse repetition rate and measured spot size of the excitation beam, and are now also highlighted on page 4 of the manuscript, as discussed in our response to Reviewer 2.

2. *In the discussion section the authors comment briefly on approaches that could be made to improve the modulation depth, and approaches to making an electrically controlled device (i.e. using piezoelectric transducers instead of the bulky amplified laser). These approaches are clearly very exciting for communications applications. Can the authors comment on the cost to modulation bandwidth on using these alternative approaches?*

The authors agree that this is a useful and interesting point to add to the manuscript. Therefore, the following discussion has been added on page 12 of the revised manuscript to provide some context to the improvements suggested:

"For example, an AlN thin-film transducer with a few volts applied could be used to generate strain amplitudes up to 10^{-4} , comparable to the strains used in our experiment and theory. Piezoelectric transducers have been demonstrated to operate up to ~ 96 GHz,⁴⁷ while SASER devices have been demonstrated to operate up to 325 GHz,⁴⁶ implying that using either of these methods as the source of the modulating strain-pulse could enable larger (although still comparable) modulation bandwidths than that achieved here."

Reviewer #2

We thank Reviewer #2 for their positive comments and careful reading of the manuscript.

1. *The experimental setup lacks some details, especially regarding the coupling of the near-ir ultrafast pulses to the back of the QCL. How large is the spot? is the whole laser modulated? if not, what is the estimated overlap? The informations are in the Methods section, I think they should be present in the setup figure 1.*

The authors agree with the reviewer's comment that the experimental setup lacked some useful details which were presented in the Methods section of the manuscript. Information on the size of the QCL device, the optical spot size and overlap between the two has now been moved from the Methods section to page 4 of the revised manuscript:

"The THz QCL device with dimensions of (2000 x 150)- μm^2 was mounted onto the cold finger of a cryostat, with a 1.0 x 0.3-mm² slit aperture allowing optical access to the aluminium transducer layer..... The optical pulses were focused using cylindrical lenses to form a beam with dimensions of ~ 1.1 mm x 0.3 mm at the QCL position, resulting in $\sim 50\%$ spatial overlap between the optical beam and the QCL device."

2. *FIG 1C, the LIV plot looks a bit weird, like the graphs are shifted with respect to the axes because the I-V curve seems not to go trough zero at zero bias. Please explain/correct*

The authors would like to thank the reviewer for noting a small error in the data presented in Figure 1(c), resulting from an increased resistance introduced to the detection circuit by the microwave bias-tee. Figure 1(c) has been corrected in the revised manuscript, and the experimental values of V_1 - V_3 correspondingly altered in the text and figure captions; the values of $V(t)$ and $L(t)$ have also been corrected in Figures 2 and 3.

3. *Fig.4 The linear fit to the data seems not to be the correct choice. There are clear oscillations both in voltage and power that produce a "steplike" behaviour. Any more comments on the origin of these steps? The assumption of linearity is maybe not fitting in the present case?*

Respectfully, the authors do not agree that 'steplike' behaviour is observed in the data shown in Figure 4. Rather, the scatter in the data points is due to experimental error in both the measured amplitude of the modulation and the optical laser power. To highlight this point the authors have now included error bars in the x-axis of Figure 4 of the revised manuscript.

Concerning the choice of a linear fit, the authors note that it is well known that the strain amplitude is directly proportional to the optical laser power [ref 31]. It is anticipated similarly that the amplitude of the modulation on the THz power and voltage should vary linearly with strain amplitude and, hence, the laser power. As such, a linear model was used to fit the data, and the authors note this provides a good fit to the experimental data with reduced chi squared values of $\chi_{red}^2=0.59$ for ΔV and $\chi_{red}^2=0.70$ for ΔL . Since no 'steplike' behaviour is expected and a linear model fits the data well, the authors suggests there is no reason to alter our statements on this in the manuscript.

4. *The speculations about the possibility to modify the QCL band structure to introduce, for example, acoustic Bragg mirrors are in my opinion a bit naive, QCL structure is extremely delicate and barrier thinning and or other modifications would have a sizeable impact on the performance of the device. Such speculations, if present in the paper, should be supported by a little bit more quantitative figures, like by how much barriers would be thinned and the period of the acoustic Bragg mirror.*

The reviewer's helpful comment on the placement of acoustic Bragg mirrors indicates that our discussion around this possibility was unclear in the manuscript. The authors' intention was that the Bragg mirrors could be grown above and below the active QCL region, rather than being incorporated within the QCL structure itself. These Bragg mirrors could be highly doped, so they would still work as a THz plasmonic waveguide. On page 11/12 of the revised manuscript we have added the following text to clarify this:

"In practice this could be accomplished, for example, using acoustic Bragg mirrors grown above and below the QCL active region stack; these Bragg mirrors could also be highly doped, allowing them to act as a THz plasmonic waveguide. The frequency of the acoustic standing wave is determined by the period of the Bragg mirrors, with a 100 GHz standing wave requiring each period of the Bragg superlattice to be ~25 nm thick."

Regarding the use of thinner barriers to increase the modulation depth, we note there are several published QCL designs that utilize thin injection barriers (e.g. scattering-injection structures), and the authors' research group has proven experience in designing and fabricating high performance QCLs with thin injection barriers [ref 45]. In the revised manuscript, a fuller discussion on devices with thin injection barriers has therefore been added to p11:

"By using thinner injection barriers (such as ~24-30 Å barriers theoretically proposed for resonant phonon depopulation structures⁴⁴, or the ~44 Å barriers used for high performance QCL devices based on scattering-injection structures⁴⁵), it is hypothesized that the induced deformation potential could result in greater perturbation of electron transport through the structure."

5. *Regarding the simulations that include the acoustic wave perturbation to the QCL potential, which value has been used for η , the strain amplitude? is the value used compatible with what produced by the laser pulses? would there be a way to assess more quantitatively the strain amplitude and characterize better the acoustic wave perturbation which impinges onto the QCL? Maybe looking at some Raman measurements could help.*

We thank the reviewer for this suggestion. On page 7 of the manuscript, it is stated that the amplitude of the perturbation caused by the acoustic wave is 0.01 to 1 meV. Taking the value of the deformation potential of ~10 eV for GaAs, this corresponds to strain amplitudes in the range 10^{-6} to 10^{-4} . The text has been modified in the revised manuscript to incorporate this information:

"The amplitude of the perturbation caused by the acoustic wave is typically ~0.01–1 meV,³⁸ corresponding to strain amplitudes in the range 10^{-6} - 10^{-4} (using the deformation potential of

GaAs of ~ 10 eV per unit strain), which is consistent with piezospectroscopic measurements using metal transducers on GaAs³⁸.”

As already noted in the manuscript on page 6, there is no evidence of acoustic nonlinearity in our experimental data, even at the highest pump fluences used (see Fig. 4 for example) which would be expected for strain amplitudes above $\sim 10^{-4}$ [ref 38]. As such the authors conclude that the upper bound on the strain amplitude in our experiment is below this value of $\sim 10^{-4}$, which is consistent with the range of strain amplitudes stated above.

The reviewer also asks if it is possible to measure the strain amplitude, potentially using Raman measurements. A more direct method would be to use the piezospectroscopic effect, where the strain perturbs slightly the wavelength of interband optical emission by a quantum well. The paper by Akimov et al. [ref 38] reports values (10^{-4} for excitation fluences of 3 mJcm^{-2}) for the strain amplitude generated using 800 nm optical excitation of a thin metal film transducer on GaAs, similar to that used in the experiment described in our manuscript. A citation to this paper has now been included on page 7 of the revised manuscript as further justification for the strain amplitudes used in our theoretical model.

6. *Still concerning the quantitative amplitude of the strain, this would be useful also to support the claims of a possible use of this effect employing on-chip solutions like piezo transducers.*

The reviewer asks if the strain amplitudes in our experiment are consistent with those that could be produced by a piezoelectric transducer, which would justify our suggestion of a possible on-chip solution using piezoelectric transducers. The answer to this is a definite yes, as can be seen by considering the example of an AlN thin-film piezoelectric transducer: the piezoelectric coefficient of AlN is $\sim 2 \text{ pm/V}$. Therefore, to obtain a strain of 10^{-4} would require an electric field of around 40 MVm^{-1} , which is below the breakdown field of AlN ($\sim 100 \text{ MVm}^{-1}$); this could be obtained by applying $\sim 10 \text{ V}$ to a 200 nm -thick transducer. The following sentence has therefore been added to page 12 of the manuscript:

“For example, an AlN thin-film transducer with a few volts applied could be used to generate strain amplitudes up to 10^{-4} , comparable to the strains used in our experiment and theory.”

7. *In the references, I think there could be space for the paper by Bruchhausen et al., Journal of Applied Physics 112, 033517 (2012), who investigated coherent acoustic phonons in THz qcls by means of pump probe spectroscopy.*

The authors thank the reviewer for this suggestion. We note this previous work did not report measurements on a working (lasing) QCL device with electrical bias applied, but instead just the unbiased layer structure, and furthermore only reported the measurements of sub-THz acoustic modes optically generated directly within the structure, rather than any THz emission. However, based on a more recent reading of the paper, the authors find the results within can be used to support our assumption that the acoustic pulse is not strongly perturbed as it propagates through the QCL structure (hence the many acoustic ‘echoes’ we observe in our data). In the revised version, we have therefore added a citation to this paper and the following text on page 4/5:

“We assume that the acoustic pulse is not significantly perturbed as it propagates through the QCL structure. This is consistent with pump-probe measurements of the acoustic phonon modes of typical QCL structures³⁴ which show phonon lifetimes less than or equal to the time for an acoustic wave to propagate once through the structure, and indicates narrow phonon stop bands in the frequency range of interest.”

To conclude, I think the paper needs more quantitative assessments on the simulations and clarifications on the previous points. The claim of applicability of these technique to modulate actual devices are in my opinion excessive and they should be turned a bit down. The bottom access of this kind of technique or others presents some serious troubles in terms of efficient heat sinking of the QCL.

The reviewer’s comment on the ‘claim of applicability of these technique to modulate actual devices’ strikes the authors as somewhat confusing, since this manuscript describes the physical experiments in which a THz QCL is modulated using an acoustic pulse, along with a theoretical description of these experimental results. As such, the authors maintain confidence in their statement that this is an applicable technique to modulate actual devices. Device modifications are also proposed for improving the modulation depth and modulation frequency. The authors trust the additional details added to the manuscript concerning these proposals (see #4 and #6 above) are satisfactory to the reviewer.

The reviewer also comments on the issue of efficient heat sinking of the QCL device. We wish to clarify that in our experiments the aperture in the copper heatsink was small (1 mm x 0.3 mm), exposing only half of the substrate to the optical pulse, and allowing the other half of the substrate to be bonded to the heatsink in the typical manner (by indium-bonding). This mounting of the QCL allowed for CW operation up to 40 K, and pulsed operation up to 100 K. For reference, with standard processing/mounting, these QCL structures were observed to lase up to 111 K in pulsed operation and up to 50 K in CW, suggesting that the heat sinking was not greatly adversely affected. As mentioned in point #1, details of the spatial overlap between the optical beam and QCL ridge have now been added to the revised manuscript. We also note that by placing the Al-transducer layer on the underside of the QCL it was possible to use time-of-flight measurements to distinguish between the effects of the acoustic pulses and any possible response of the QCL device to light. This would have been more difficult had we chosen to generate the acoustic pulse from the top contact of the QCL ridge. However, in a practical application, it would be perfectly feasible to position a transducer on top of the QCL ridge.

Reviewer #3

We would like to thank Reviewer #3 for their comments, and their recognition that demonstration of the physics is the most important aspect of this manuscript, and that additional work will be needed in the future to explore the full range of potential applications.

1. *The authors claim they demonstrate a new mechanism. I would say that at least two references probably forgotten by the authors already reported acousto-electric effects or transient built-in field leading to the modulated/transient THz emission. The main difference in the submitted paper comes for the system in which such acousto-electric effect takes place. In the references cited below, such acousto-electric effect was demonstrated in semiconductors hetero-structures (Armstrong et al). In Van Capel et al's paper, semiconductor superlattices were studied ; the acousto-electric effect is not directly discussed but the physics is very close even if in the latter one, piezoelectric effect comes in addition.*

References :

Armstrong et al, Observation of terahertz radiation coherently generated by acoustic waves, NATURE PHYSICS VOL 5 285-288 (2009)

Van Capel et al, Correlated terahertz acoustic and electromagnetic emission in dynamically screened, InGaN/GaN quantum wells PHYSICAL REVIEW B 84, 085317 (2011)

I strongly encourage the author to cite/comment this work to make the survey more complete. Moreover, they also should then demonstrate more clearly the new opportunities that their experiments provide for the THz modulation point of view.

We thank the reviewer for these suggestions. Firstly, we would like to stress that the 'new mechanism' we claimed was in relation to a new mechanism to modulate the THz emission from a device, rather than a new physical process through which this modulation was occurring. This is now stated explicitly in the abstract of the revised manuscript:

"Here, we demonstrate a new mechanism to modulate the emission from a QCL device..."

The authors agree that the Armstrong paper has some relevance to our work in that it concerns generating THz radiation using acoustic waves. A sentence has therefore been added on page 11 of the revised manuscript citing the Armstrong paper:

"The authors note that in experiments demonstrating direct acoustic-to-THz conversion the effect is weak compared to emission from a QCL,⁴³ and that even the 6% modulation demonstrated here is considerably greater than the total emission from these direct acoustic-to-THz conversions."

The Van Capel paper, however, documents the concurrent release of THz electromagnetic radiation and THz acoustic phonons due to the dynamical screening of the internal electric field in InGaN/GaN multiple quantum wells, and does not discuss the conversion of the acoustic wave to THz radiation at all. As such, we do not feel it would be relevant to cite this paper within our work since we do not feel this approach provides new opportunities for the modulation of THz radiation.

2. *I do not understand the advantage of using acoustic waves when the THz modulation last around 6ns (166MHz), i.e. an order of magnitude slower than what can be achieved with electrical modulation (Ref 19). Could the authors comment this point?*

We wish to clarify for the reviewer that the 6 ns duration of the modulation observed in our experiment corresponds to the round-trip transit time of the acoustic wave through the QCL active region as stated in the manuscript on page 5. As is also stated in the manuscript, this duration is not indicative of the timescales of the underlying acoustoelectric processes; owing to the short (~ps) carrier lifetimes in QCLs, a strain-induced deformation of the bandstructure will cause perturbations to the electrical carrier transport on ultrafast timescales. This is confirmed through our theoretical model, which shows this modulation speed is inherently limited only by the transit time of the acoustic pulse through an individual injection region of the heterostructure stack. On page 10 of the manuscript, we propose device modifications that could enable modulation frequencies exceeding ~200 GHz to be achieved. Demonstration of these faster modulation speeds is the next logical step to this work.

3. *Regarding the application, why the authors did not study a mid-infrared QCL working at room temperature to investigate this acoustoelectric effect?*

We are pleased to clarify that there are two reasons for this; a) it was necessary to cool the device below 50 K in order to make the substrate transparent to the acoustic phonons and b) the QCLs available to the authors were THz frequency devices, not mid-IR. In principal the authors believe modulation of a mid-IR QCL at temperatures below 50 K could be feasible.

4. *It is not clearly explained how a 20ps acoustic pulse becomes a bipolar 6ns pulse in the structure. Page 5, the authors say that the “bipolar” shape of the THz “pulse “ (fig 2) is due to the existence of an acoustic reflection. The L curve (black curve) clearly exhibits a change of sign which is not the case of V. This part is not convincing. Maybe the time window of the inset of Fig2 should be increased to evidence more clearly the change of sign?*

We believe the reviewer is conflating the duration of the acoustic pulse with the duration of the measured voltage perturbation. In our experiments the duration of the measured voltage perturbation is determined by the ~6 ns transit time of the acoustic pulse (which has a duration ~20 ps) through the QCL heterostructure (see point #2). This is stated on page 5 of the manuscript, while discussing the data in Figure 2, and also stated in the Discussion section when suggesting methods for improving the maximum modulation speed. In order to clarify this point and to help avoid future confusion, the text at the top of page 5 of the manuscript has been revised as follows:

“Although the acoustic pulse has a duration of ~20 ps, the duration of the acoustically-induced effect is determined by the round-trip transit time of the acoustic wave through the 13.9- μm -thick QCL active region, which was calculated to be ~6 ns using the speed of sound in GaAs. This is shown experimentally in the inset of Fig. 2.”

We further wish to clarify for the reviewer that although the acoustic pulse is itself bipolar, the voltage and power data displayed in Figures 2 and 3 is a consequence of the transit of the acoustic pulse through the active region and is not therefore inherently bipolar. When

discussing the acoustic reflection from the top contact of the QCL and its effect on $V(t)$, we state carefully in the manuscript text that the voltage modulations “comprise an initial ~ 3 -ns pulse, followed by a ~ 3 -ns-long shoulder” but not that it is ‘bipolar’ in nature. In fact it is mentioned several times in the experimental and theoretical discussion that the voltage modulation is always positive. The modulation in $L(t)$ presented in Figures 2 and 3 may (coincidentally) appear to be bipolar, but the signal is distorted by ringing due to resonance in the electrical circuit (as discussed on pages 5 and 6 of the manuscript). The revised manuscript has been modified to make the unipolar-nature of both modulations clear with the following text on page 6:

“It is important to note that both the voltage and power modulations are unipolar in nature, although distortion by ringing due to resonance in the electrical circuit does cause the modulation in $L(t)$ to appear bipolar at certain biasing conditions. However, as is evident in Fig. 3 (in which the shaded areas indicate the times at which the acoustic pulse will be acting on the QCL active region), it can be seen that these ringing effects occur after the acoustic pulse has left the QCL ridge. As such, these effects are attributed to the active region relaxing to its unperturbed state, and not attributed directly to the modulation due to the passage of the acoustic pulse through the active region.”

5. *The authors say that increasing V_3 gives rise to a decrease of L and V_1 the converse effect. I see the first effect but not the converse one. Why?*

The authors state the acoustic pulse is bipolar, but due to electronic diffusion, this should not be the case (see Ref 40). So, could this asymmetry be responsible for an asymmetry in the phonon-induced tunneling (reverse/forward) in the sense that the compressive and expansion strain tails do not have the same spatial extension and then the deformation-potential perturbation does not have the same spectrum (i.e. not the same frequency components) ? This should lead to a different rate of transition in the time-dependent Schrödinger equation.

As shown in our work, the magnitudes of the modulations in the voltage and power are dependent on the QCL bias, with larger modulations occurring at bias V_3 than at V_1 . At lower biases (such as at V_1) the modulated signal is affected more by noise in the experimental system. Nevertheless, it is observed that the acoustic modulation in the power signal is consistently positive in the time window where the acoustic modulation occurs. Specifically, as shown in Figure 3, an increase ΔL of approximately $\sim +0.1$ mW can be seen in $L(t)$ at V_1 which occurs at the same time as the decreases of ~ -0.25 mW at both V_2 and V_3 . A sentence clarifying this has been added on page 10 of the revised manuscript:

“At lower biases, the modulated signal is affected more by noise in the experimental system, making it more difficult to determine the effect of the acoustic pulse on $L(t)$. Nevertheless, it was observed that the modulation of power is consistently positive in the time window where the acoustic modulation occurs.”

In response to the second comment, we note it is generally accepted that the strain pulse has a bipolar shape when generated by optical excitation of a metal film (such as aluminium in this case) [see ref 31]. Ref. 42 (previously ref 40) refers to generation directly in a

semiconductor surface, and does not describe the generation approach in our experiment. Strain pulses in the nonlinear regime would also result in asymmetric pulse shapes. However, as it is clear that this experiment operated in the linear regime, the compressive and expansion strain tails should not be significantly different.

6. *The authors calculate the acoustically-induced tunnelling (injection level towards the upper level). There is no discussion regarding the perturbation of the dipolar matrix element involved in the radiative recombination? Why?*

The acoustic perturbation is small (< 1 meV), and is certainly much lower than either the conduction-band offsets in the QCL or the energy separation between the lasing levels. This means that the perturbative effect on the wave functions, and hence dipole matrix element, will be extremely small, consistent with a first-order perturbation approximation.

Theoretically though, the change in the dipole matrix element would be extremely difficult to calculate with any accuracy, as the perturbation varies on a similar time-scale to the state lifetimes. As such, a time-independent model would be inappropriate. Conversely, a TDP model (as used in our manuscript) produces a tunnelling rate directly without determining any changes to the wave functions. Furthermore, the authors note that the peak change in tunnelling rate predicted from the model is on the order of 40%, which would have a more significant effect on the laser gain than any change in the dipole matrix element since the energy separation of lasing levels is ~ 12 meV. The calculation of the interaction probability between the lasing states yielded a 10^{-3} % effect, which is considerably lower than the 40% change in tunnelling rate happening between the resonant states. A comment discussing this has been added to page 8 of the revised manuscript:

“The time-dependent net tunnelling probabilities between the injector and upper lasing states were found to be perturbed by up to $\sim 40\%$ by the presence of the acoustic wave; for comparison, the same calculation between the upper and lower lasing states yielded a $\sim 10^{-3}\%$ effect.”

REVIEWERS' COMMENTS:

Reviewer #1 (Remarks to the Author):

The authors have addressed all my comments well and I feel that the work should be published in its current form.

Reviewer #2 (Remarks to the Author):

The Authors have answered satisfactorily to my queries and the paper now can be accepted for publication

Reviewer #3 (Remarks to the Author):

The authors have significantly improved the manuscript by addressing most of the issues. The authors have provided additional details (both theoretical and experimental) to make the text clearer and more convincing. In particular, the modulation rate of 6% is better explained/supported by now. The QCL is a complex heterostructure (even developed at the semi-industrial level) and studying the interaction of carriers with coherent acoustic phonons requires some advanced modeling. However, at this stage, the authors finally propose a reasonable description of the THz emission modulation effect with reasonable assumptions. I think this paper will be of interest for a broad community of solid state physics including the physics of QCL and the physics of coherent phonon. I think the paper is suitable for publication.